# Effect of High Pressure on the Solidification of Al–Ni Alloy

**Xiao-Hong Wang** [1,*], **Duo Dong** [1] and **Xiao-Hong Yang** [2]

1   Key Laboratory of Air-Driven Equipment Technology of Zhejiang Province, Quzhou University, Quzhou 324000, China; dongduo@qzc.edu.cn
2   School of Mechanical and Electrical Engineering, Jinhua Polytechnic, Jinhua 321017, China; 20050626@jhc.edu.cn
*   Correspondence: qu_wxh36098@qzc.edu.cn

**Abstract:** The effect of high pressure on the microstructure of hypo-peritectic Al–38wt.%Ni alloy was studied. The results show that $Al_3Ni$ and $Al_3Ni_2$ phases coexist at ambient pressure. However, it becomes a typical hyper-eutectic microstructure when synthesized at 2 GPa and 4 GPa. Meanwhile, the interface temperature of $Al_3Ni$ and $Al_3Ni_2$ phases was calculated with the combination of the BCT dendrite growth model, which is suitable for the $Al_3Ni_2$ phase. According to the highest interface temperature principle, the result shows that the $Al_3Ni$ phase dominates over 1–5 GPa. Finally, the Debye temperature and potential energy of the hypo-peritectic Al–38wt.%Ni alloy under different pressures were researched. Based on the low temperature specific heat-capacity curve. The Debye temperatures at ambient pressure, 2 GPa, and 4 GPa are 504.4 K, 508.71 K and 515.36 K, respectively, and the potential energy in the lowest point decreases with the increase of pressure.

**Keywords:** Al–Ni alloy; high pressure; phase selection; potential energy; Debye temperature

## 1. Introduction

Intermetallic compounds have attracted more and more attention due to their superior heat resistance, high specific strength, high thermal conductivity and high oxidation resistance, as well as superior toughness and good formability to ceramic materials [1–3]. While among these intermetallic compounds, the Al–Ni series is one of the most important and promising candidate high-temperature materials [4,5]. There are six kinds of intermetallic compounds in the Al–Ni alloy: $Al_3Ni_2$, $Al_3Ni$, $Al_4Ni_3$, $AlNi$, $Al_3Ni_5$ and $AlNi_3$ [6,7]. It can be divided into two categories according to their solid solubility: (1) intermetallic compounds with high solid solubility, such as $AlNi$ and $Al_3Ni_2$; (2) intermetallic compounds with low solid solubility or even nil solid solubility, such as $Al_3Ni$. The solidification and growth characteristics of intermetallic compounds with a certain range of solid solubility can refer to the solidification of solid solution, but for intermetallic compounds with nil solid solubility (the composition does not change with the change of temperature and pressure, and the equilibrium distribution coefficient is 0), whether the constitutional undercooling criterion, dendrite tip radius, interface stability growth equation and minimum undercooling theory based on solid solution characteristics are suitable for the solidification research of nil solid solution intermetallic compound $Al_3Ni$ and whether the final solidification microstructure will change greatly are all worth considering. Based on a phase diagram of the Al–Ni alloy, the solidification process of peritectic composition is rich in the intermetallic phase. Therefore, the Al–Ni alloy with hypo-peritectic composition was selected.

During the equilibrium solidification of peritectic alloys, the primary phase is metastable compared with the peritectic phase. However, under nonequilibrium conditions, including high pressure solidification, competitive nucleation and growth may occur between peritectic and primary phases [8,9]. Zhai et al. [10] reported that once the undercooling of the Cu–70wt.%Sn alloy exceeds the critical value of 220 K, the peritectic phase precipitates

directly from the metastable liquid while the nucleation of the primary phase is suppressed. An enrichment microstructure [11] can be formed by the different growth behavior of the metastable and the stable phase. Kubin and Estrin [12] and Caroli [13] pointed out that the phase with the highest interface temperature preferentially grows, which constitutes the main content of the highest interfacial growth temperature criterion. Based on the criterion, Fredriksson [14], Gill and Kurz [15], and Umeda [16] studied the competition behavior between stable and metastable phases under directional solidification. Vandyoussefi [17] successfully applied it to predict the phase selection of the Fe–Ni peritectic alloy under rapid solidification. The highest interface temperature criterion has a great relationship with the dendrite tip radius. However, for classical dendrite growth theory, for example, BCT theory [18], the derivation of dendrite tip radius is based on the neglect of the kinetic undercooling [19,20]. Nevertheless, for the intermetallic compound phase with nil solubility, the situation may be totally different, and it is rarely investigated [21].

Pressure is an important thermodynamic parameter that can greatly enhance the kinetic undercooling. Meanwhile, during the solidification process, it can also affect the phase formation. In terms of solidification microstructure, Ma et al. [22] studied the solidification microstructure of the Al–20wt.%Si alloy under different pressures. The results showed that the primary Si phase exists after solidification at atmospheric pressure and 1 GPa, but disappears at higher pressure. The microstructure of the Al–xCu (x = 15, 33, and 40wt.%) alloy solidified under different pressures was studied by Liu et al. [23]. The results showed that the eutectic $Al_2Cu$ phase varied from dendritic to spherical morphology in the Al–15wt.%Cu alloy with the application of pressure. Both the secondary arm spacing in the Al–33Cu alloy and the size of primary $Al_2Cu$ in the Al–40wt.%Cu alloy had decreased under high-pressure solidification. The Ti– 48at.%Al peritectic alloy solidified at different pressures was studied by Wang et al. [24]. The results demonstrated that the phase transformation $L+\beta \rightarrow \alpha$ is greatly inhibited. While in terms of solidification theory, the application of the curvature radius equation obtained by Huang et al. [25] in the pressurized solidification process to the Al–Si system demonstrated that the partial derivative with respect to pressure is positive. Meanwhile, the effect of pressure on the growth velocity was also developed by Huang et al. [25]; the result proved that the growth velocity decreases in the pressurized process when the volume increases during melting. Emuna [26] and Zhou [27] investigated the pressure dependence of the liquidus temperature and the eutectic point of some binary systems.

Debye temperature is an important physical property of metal materials; it is a thermoparameter to describe many physical phenomena of solid-state materials associated with the lattice vibrations [28]. It depends on properties such as elastic constants, lattice thermal conductivity, and specific heat [29]. Theoretically, Lindemann [30] suggested a relationship between Debye temperature and melting temperature based on the function of Einstein expressing the variation of the atomic heat of various elements, while Daoud [31] established an empirical expression about Debye temperature and bond length, and Abrahams et al. [32] studied the relationships between Debye temperature and compressibility, microhardness, etc.

In the present work, the effect of high pressure on the microstructure evolution and the phase selection of the Al–38wt.% Ni hypo-peritectic alloy is calculated and investigated first as it contains both intermetallic compounds and peritectic reaction. Then the effect of high pressure on the potential energy and Debye temperature are researched.

## 2. Materials and Methods

Using pure Aluminum (Beijing Xing Rong Yuan Technology Co., Ltd., Beijing, China, 99.99%) and Nickel (Beijing Xing Rong Yuan Technology Co., Ltd. 99.99%) as raw materials, 38wt.% Al–Ni alloy was prepared by a button ingot melting furnace. The total weight of each button ingot was no more than 50 g, and the diameter is about 40 mm. It was repeatedly melted 5 times with proper current and power. The high-pressure solidification sample is a cylinder with a diameter of 6 mm and a length of 8 mm. The experiment

was carried out on a 700-ton multi-anvil device (self-assembled, Japan). The Bi phase transition was used as the fixed point to determine the pressure generated, and inserting hot junction of thermocouple (R-type) in the center of a heater was used to calibrate the temperature. The pressure was set at 2 GPa and 4 GPa. The sample was melted in a graphite tube furnace. Pyrophyllite was used as encapsulant and pressure-transmitting medium. When the pressure rose to the target value, the sample was kept at the target pressure and temperature for 1 hour to ensure complete melting. Then the current decreases rapidly to 0, stop heating and the sample is cooled and solidified. Finally, samples were taken out for analysis after pressure relief. The morphology of the samples was obtained by Scanning Electron Microscopy (SEM; Hitachi s-8100, Tokyo, Japan), and the composition was analyzed by Energy Dispersive X-Ray (EDS, Hitachi su8010, Tokyo, Japan). The low-temperature heat-capacity performance was tested with Physical Property Measurement System (PPMS-9, Quantum Design, San Diego, CA, USA).

## 3. Results

*The Stable Growing Wavelength of Intermetallic Compound Al$_3$Ni with Nil Solid Solubility*

The solute distribution equation at the frontier of Al$_3$Ni interface deduced by Liu [21] is shown below:

$$C_L = C_0 + (C_L^* - C_0) \exp\left(-\frac{V_z'}{D}\right) \tag{1}$$

The meaning of the parameters is shown in Table 1.

**Table 1.** The physical parameters used in part 3 and 4.

| Physical Quantities | Symbol (unit) |
| --- | :---: |
| $z'$—distance to the solid-liquid (S-L) interface | m |
| $z$—distance to the beginning of solidification | m |
| $D$—diffusion coefficient | m$^2$/s |
| $V$—solidification velocity | m/s |
| $C_0$—initial alloy concentration | at.% |
| $C_L$—The solute distribution at the frontier of Al$_3$Ni interface | at.% |
| $C_L^*$—liquid composition at S- L interface | at.% |
| $C_\beta$—Al$_3$Ni concentration | at.% |
| $T_f$—melting temperature | K |
| $G$—actual temperature gradient in front of S-L interface | K/m |
| $\Gamma$—Gibbs Thompson coefficient | K·m |
| $V_\phi$—velocity of the perturbation interface | m/s |
| $a$—thermal diffusivity | m$^2$/s |
| $K_\phi$—curvature | /m |
| $\Delta H_f$—latent heat of the metal | J/mol |
| $V_0$—sound speed in the liquid | m/s |
| $R_g$—gas constant | J/mol·K |
| $\Delta h_f$—latent heat of fusion | J/m$^3$ |
| $c$—volumetric specific heat | J/m$^3$·K |
| $R$—radius of the dendrite tip | m |
| $\Delta T$—the undercooling of the dendrite tip | K |
| $T_L$—liquidus temperature | K |
| $\Delta T_t$—heat undercooling | K |
| $\Delta T_c$—solute undercooling | K |
| $\Delta T_r$—curvature undercooling | K |
| $\Delta T_k$—kinetic undercooling | K |
| $m_p$—liquid slop under high pressure | K/at.% |
| $\sigma^*$—stability constant = $1/4\pi^2$ | - |
| $\xi_t$—solute stability parameter | - |
| $\xi_c$—solute stability parameter | - |
| $I_v(Pc)$—solutal Ivantsov function | - |
| $P_t$—thermal Péclet number = $VR/2a$ | - |
| $P_c$—solute Péclet number = $VR/2D$ | - |
| $\mu$-kinetic coefficient | - |
| $\xi_l$—function related to Péclet number | - |
| $\xi_s$—function related to Péclet number | - |

According to the mass balance law, the solute distribution at the frontier of Al$_3$Ni is as below [21]:

$$C_L^* = (C_0 - C_\beta)\frac{V}{D}z + C_0)$$ (2)

The result shows that, when $C_0 < C_\beta$, with the increase of $z$, $C_L^*$ decreases linearly. Equation (1) also displays that a steady boundary layer cannot be established during the growth of the Al$_3$Ni planar interface; it is completely different from solid solution.

When $m > 0$ and $k > 1$, the temperature distribution at the frontier of solid liquid interface can be expressed as follows:

$$T_L = T_f + m(C_0 - C_\beta)\frac{V}{D}z \cdot \exp\left(-\frac{V_z}{D}\right)$$ (3)

Then, the temperature gradient equation is as follows:

$$\frac{dT_L}{dz} = m(C_0 - C_\beta)\frac{V}{D}\exp\left(-\frac{V_Z}{D}\right) - m(C_0 - C_\beta)\frac{V^2}{D^2}z \cdot \exp\left(-\frac{V_z}{D}\right)$$ (4)

When $z \to 0$ and $e_z \approx 1 + z$. Equation (4) can be simplified as:

$$\frac{dT_L}{dz} = m(C_0 - C_\beta)\frac{V}{D}\left(1 - \frac{V_z}{D}\right)^2$$ (5)

Therefore, it can be deduced that the constitutional undercooling condition of Al$_3$Ni phase is:

$$G \leq m(C_0 - C_\beta)\frac{V}{D}\left(1 - \frac{V_z}{D}\right)^2$$ (6)

According to Equation (6), when $C_0 \leq C_\beta$, there is no constitutional undercooling.

It is assumed that the perturbation has infinitesimal amplitude and it does not affect the thermal and solute concentration fields. The perturbed interface is represented by a simple sinusoidal function [33]. For the Al$_3$Ni phase, the interface temperature $T_\phi$ can be derived from the assumption of local equilibrium:

$$T_\phi = T_f + m\Delta C - \frac{V_\phi}{\mu} - \Gamma K_\phi$$ (7)

$$T_\phi = T_f - m(C_0 - C^*) - \frac{\Gamma V_\phi}{\mu} - \Gamma K_\phi$$ (8)

It shows that the difference between interface temperature and melting point is equal to the sum of the temperature caused by solute redistribution, kinetic undercooling, and curvature undercooling.

Thus, the interface temperature at the peak ($t$) and the valley ($d$) can be expressed as:

$$T_t = T_f - m(C_0 - C_t) - \frac{V_\phi}{\mu_t} - \Gamma K_t$$ (9)

$$T_d = T_f - m(C_0 - C_d) - \frac{V_\phi}{\mu_d} - \Gamma K_d$$ (10)

Then, the difference between the peak and valley interface temperatures can be written as:

$$T_t - T_d = m(C_t - C_d) - \left(\frac{V_\phi}{\mu_t} - \frac{V_\phi}{\mu_d}\right) - \Gamma(K_t - K_d)$$ (11)

Meanwhile, the corresponding kinetic coefficient can be written as follows [33,34]:

$$\mu_t = V_0\Delta H_f / R_g T_f T_t$$ (12)

$$\mu_d = V_0 \Delta H_f / R_g T_f T_d \tag{13}$$

And the curvature of the peak and valley can be determined by evaluating the second derivative of the interface shape function at $y = \lambda/4$, $y = 3\lambda/4$:

$$K_t = -K_d = 4\pi^2 \varepsilon / \lambda^2 \tag{14}$$

Since $T_t - T_d = -2\varepsilon G$, $C_t - C_d = 2\varepsilon G_c$.

Therefore, substituting Equations (11)–(13) into Equation (14), Equation (15) can be obtained:

$$R = \lambda = 2\pi \left( \frac{\Gamma}{mG_c - G\left( \frac{V_\phi R_g T_f}{V_0 \Delta H_f} + 1 \right)} \right)^{1/2} \tag{15}$$

Table 2 is the physical parameters used for the calculation of the dendrite tip radius of Al$_3$Ni during the high-pressure solidification of the Al–38wt.%Ni alloy. Considering the parameters that can be affected by pressure, such as solidification velocity *V*, diffusion coefficient *D*, interface energy $\sigma$, etc. The effect of pressure on the dendrite tip radius of the Al$_3$Ni phase in the Al–38wt.%Ni alloy is shown in Figure 1. It demonstrates that the dendrite tip radius increases with the increase in pressure.

**Table 2.** The physical parameters used for calculation [34].

| Physical Quantities | Symbol (unit) | Al–38wt.%Ni |
|---|---|---|
| Liquidus temperature | $T_L$ (K) | 1240 |
| Density (Al$_{solid}$) [a] | $\rho_{SAl}$ (g/cm$^3$) | 2.7 |
| Density (Al$_{liquid}$) [a] | $\rho_{LAl}$ (g/cm$^3$) | 2.375 |
| Density (Ni$_{solid}$) [a] | $\rho_{SNi}$ (g/cm$^3$) | 8.908 |
| Density (Ni$_{liquid}$) [a] | $\rho_{LNi}$ (g/cm$^3$) | 7.81 |
| Thermal conductivity of Al [a] | $\kappa_{Al}$ (W/m·K) | 235 |
| Thermal conductivity of Ni [a] | $\kappa_{Ni}$ (W/m·K) | 91 |
| Mole specific heat | $C_P$ (J/mol·K) | 29.3 |
| Liquidus slope of β | $m_{L\beta}$ (K/%) | 18.5 |
| Diffusion coefficient | $D_L$ (m$^2$/s) | $1 \times 10^{-7} \exp(-6.02 \times 10^{-20}/k_B \cdot T)$ [35] |

[a] Indicates that ideal mixing was assumed and that the values were obtained based on the atomic composition of the alloy.

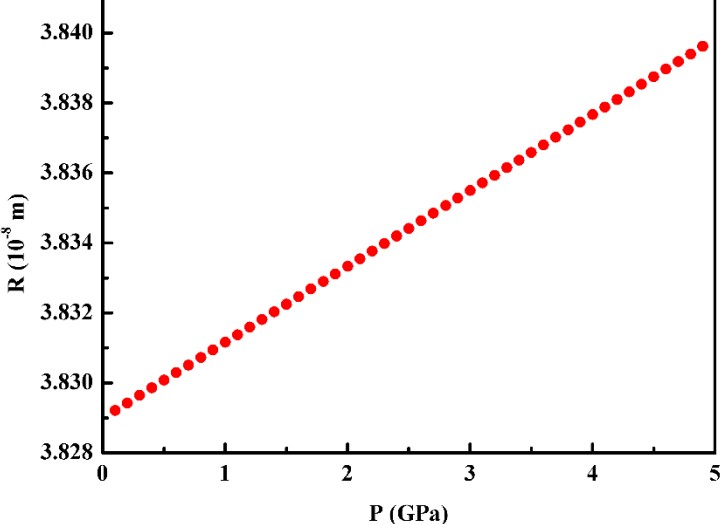

**Figure 1.** The effect of pressure on the dendrite tip radius of Al$_3$Ni during the solidification of the hyper-peritectic Al–Ni alloy.

## 4. Phase Selection under High Pressure

According to the equilibrium phase diagram of the Al–Ni alloy, the solidification of Al–38wt.%Ni starts with the nucleation of the $Al_3Ni_2$ phase from the metastable liquid. When the alloy cools down to peritectic and eutectic temperature, the reaction of $L+Al_3Ni_2 \rightarrow Al_3Ni$ and $L \rightarrow \alpha-Al+Al_3Ni$ will happen. At the end of solidification, the peritectic $Al_3Ni$ phase and eutectic phase coexist. However, the equilibrium solidification conditions are rarely achieved, which leads to the residue of the primary phase. The Back-Scattered Electron (BSE) microstructure of the Al–38wt.%Ni alloy solidified under ambient pressure is shown in Figure 2a,b. It can be seen that the whole microstructure is composed of an irregular gray massive phase, interphase eutectic phase and white small-block phase embedded in gray phase.

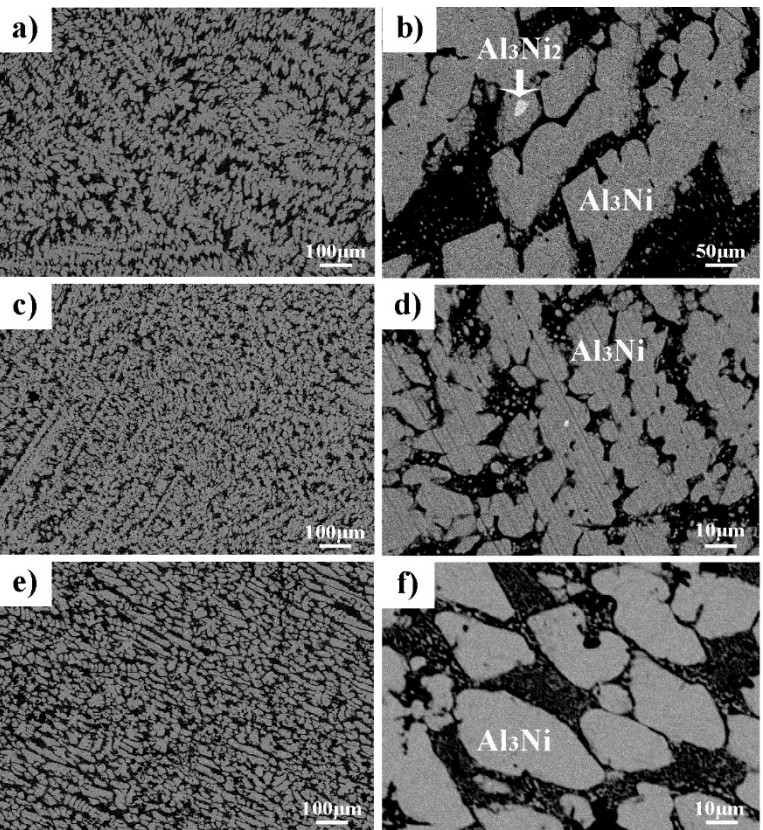

**Figure 2.** The microstructure of Al–38wt.%Ni solidified under different pressures (**a–b**) ambient pressure; (**c–d**) 2 GPa; (**e–f**) 4 GPa.

The phase composition of the Al–38wt.%Ni alloy solidified at different pressures is shown in Figure 3. It demonstrates that the phase composition of the Al–38wt.%Ni alloy solidified at ambient pressure is $\alpha$–Al, $Al_3Ni_2$ and $Al_3Ni$, while the diffraction peak of $Al_3Ni_2$ disappears when solidified at high pressures. Meanwhile, the EDS analysis of each phase shows that the content of Ni in the bright phase is 39.17at.% and in the gray phase is 25.01at.%, which demonstrates that the white and gray phase is $Al_3Ni_2$ and $Al_3Ni$, respectively. Some $Al_3Ni$ phases have obvious edges and corners, which indicates that there are two growth behaviors: faceted and nonfaceted growth. Figure 2c,d shows that the microstructure of the Al–38wt.%Ni alloy solidified at 2 GPa is a typical hypereutectic microstructure. Under the condition of high-pressure solidification, the $Al_3Ni$ phase growth is in a nonfaceted pattern and the size is about 10.36 μm. Figure 2e,f is the microstructure of the Al–38wt.%Ni alloy solidified at 4 GPa, which remains almost unchanged comparing it with the microstructure in Figure 2c,d. However, the size of

the primary Al₃Ni phase increases to 22.86 μm compared with that of 2 GPa. The results correspond to the calculation results of Figure 1.

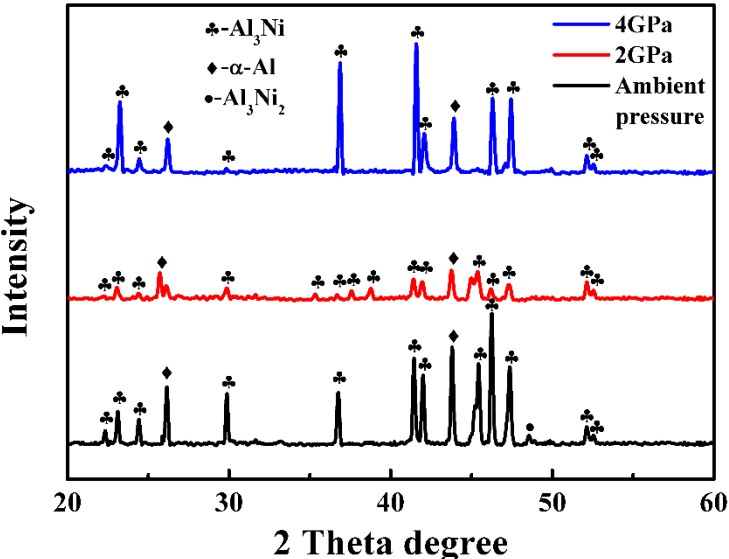

**Figure 3.** The phase composition of Al–38wt.%Ni alloys solidified at different pressures.

To explore the solidification process of the Al–38wt.%Ni alloy under different pressures, the highest interface temperature theory should be used to detect the phase selection. When the microstructure is dendrite during solidification, the interface temperature of the dendrite tip can be expressed as [33]:

$$T_d = T_L - \Delta T \tag{16}$$

Then,

$$\Delta T_{Al_3Ni} = \Delta T_r + \Delta T_t + \Delta T_k \tag{17}$$

$$\Delta T_{Al_3Ni_2} = \Delta T_c + \Delta T_r + \Delta T_t + \Delta T_k \tag{18}$$

They can be expressed as follows:

$$\Delta T_c = m_p C_0 \left[ 1 - \frac{1}{(1 - I_v(P_c)(1 - k_p))} \right] \tag{19}$$

$$\Delta T_t = \frac{\Delta h_f}{c} I(P_t) \tag{20}$$

$$\Delta T_r = \frac{2\Gamma}{R} \tag{21}$$

$$\Delta T_k = \frac{U}{\mu} \tag{22}$$

According to the dendrite growth model of BCT, the marginally stable growth wavelength of perturbations at the solid–liquid interface for the growth of Al₃Ni₂ dendrite is shown as follow:

$$R = \frac{\Gamma/\sigma^*}{\frac{P_t \Delta h_f}{c} \xi_t - \frac{2P_c C_0 (1 - K_p) m_p}{1 - (1 - k_p) I_v(P_c)} \xi_c} \tag{23}$$

$$\xi_t = 1 - \frac{1}{\sqrt{1 + \frac{1}{\sigma^* P_t^2}}} \tag{24}$$

$$\xi_c = 1 + \frac{2k_p}{1 - 2k_p - \sqrt{1 + \frac{1}{\sigma^* P_c^2}}} \tag{25}$$

Hence, the equation for the calculation of the interface temperature of the dendrite tip of $Al_3Ni_2$ and $Al_3Ni$ phase in hypo-peritectic Al–38wt.%Ni alloy is BCT model and Equation (16), respectively.

Figure 4 shows the interface temperature of the β–$Al_3Ni$ and $Al_3Ni_2$ phase, which are evaluated by inserting the physical parameters in Table 1 into Equations (15) and (17)–(25), and with the consideration of the effect of high pressure on the solute diffusion coefficient [25], equilibrium distribution coefficient [25], and entropy of fusion [8], and interface energy [8], etc. It shows that the interface temperature of the $Al_3Ni$ phase under pressure is higher than that of the $Al_3Ni_2$ phase, i.e., no metastable phase $Al_3Ni_2$ exists during the solidification of the Al–38wt.%Ni alloy under high pressure over 1–5 GPa.

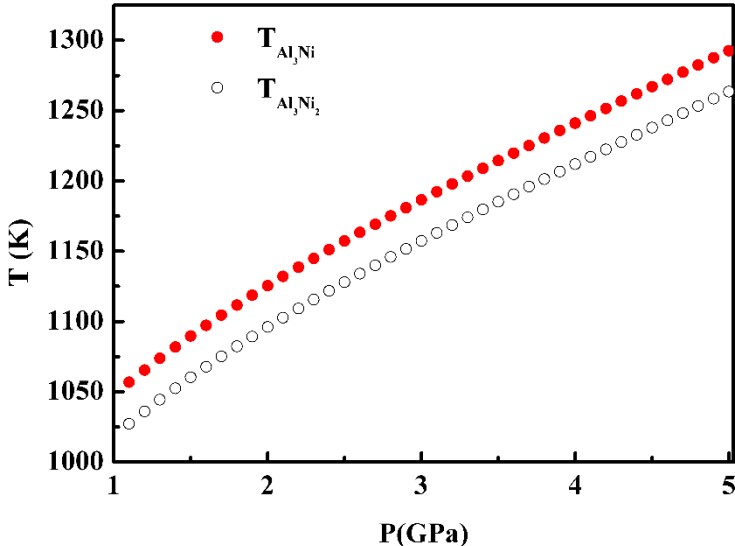

**Figure 4.** The effect of high pressure on the interface temperature of the $Al_3Ni$ and $Al_3Ni_2$ phase.

## 5. Potential Energy of Hypo-Eutectic Al–Ni Alloy at Different Pressures

The potential energy is closely related to the binding energy, which is associated with the cohesive force, Debye temperature, melting point, and activation energy, etc. Furthermore, from the above analysis, it can be seen that the pressure has an effect on the solidification parameters, such as activation energy, velocity, diffusion coefficient, etc., which has also been found to be connected with the activation energy. Therefore, it is a great necessity to calculate the potential energy and use it to predict the influence of pressure on the physical and mechanical properties of the alloy. In this section, the potential energy curves of Al–Ni alloys with different compositions under different pressures are calculated by using the potential function developed by Xie [36]:

$$W(r) = E_c\left[-n\left(\frac{r_0}{r}\right)^3 + (n-1)\left(\frac{r_0}{r}\right)^{nx/(n-1)}\right] \tag{26}$$

where $E_c$ is the binding energy, $W(r)$ is the potential energy, $r_0$ is the shortest bond length in the crystal, $x$, n are the coefficients of the potential energy function.

$n$ and $x$ can be solved by the following equations:

$$K = \frac{1}{6}\left(\frac{2n-1}{n-1}x + 3\right) \tag{27}$$

$$Q = V_0 B/\gamma \tag{28}$$

$$\gamma = \left(\frac{1}{6}\right)\left[\frac{n}{(n-1)}\right]x + \left(\frac{1}{3}\right) \tag{29}$$

$$B = \frac{4\Theta_D^2 K_B^2 r_0^2 m \cdot (1-2\varepsilon)}{9V_0 \hbar^2 j^2}, \hbar = \frac{h}{2\pi} \tag{30}$$

$$x^2 = \frac{4\Theta_D^2 K_B^2 r_0^2 m}{\hbar^2 E_c} \cdot \frac{1}{j^2}\left(\frac{n-1}{n}\right) \tag{31}$$

where $B$ is the bulk modulus of elasticity (GPa), $\varepsilon$ is the strain (0 or 0.01), $m$ is the average atomic mass (kg), $V_0$ is the molar volume (m$^3$/mol), $K$ and $Q$ are constants related to the thermal expansion coefficient.

According to the Gruneisen equation [37], the relation between the thermal expansion coefficient and specific heat is shown below:

$$a = \frac{C_V}{3Q\left[1 - K\left(\frac{U}{Q}\right)\right]^2} \tag{32}$$

where $a$ is the thermal expansion coefficient (1/K), $C_V$ is the heat capacities (J/mol·K), $U$ is the lattice vibration energies (J/mol).

The relation between $C_V$, $U$ and Debye temperature is shown as below:

$$C_v = 3R\left[12\left(\frac{T}{\Theta_D}\right)^3 \int_0^{\Theta_D} \frac{y^3 dy}{e^y - 1} - 3 \cdot \frac{\Theta_D/T}{\exp\left(\frac{\Theta_D}{T}\right) - 1}\right] \tag{33}$$

$$U = \int_0^T C_V dT \tag{34}$$

where $y = h\omega/K_B \cdot T$, $\omega$ is the lattice vibrational frequency, $\Theta_D$ is the Debye temperature (K), $R$ is the gas constant.

Combined with the linear thermal expansion coefficient, bulk modulus of elasticity, Debye temperature and the newly developed potential energy function, $n$ and $x$ can be obtained first. Then, the relationship between the potential $W(r)$ and $r$ can be obtained.

To solve the potential energy function, the Debye temperature of hypo-peritectic Al–38wt.%Ni alloys synthesized under different pressures is studied and calculated at the first place. The low temperature heat capacity of the Al–Ni alloy in the temperature range of 2–300 K prepared under different pressures is shown in Figure 5. The results show that the specific heat increases with the increase of temperature. The insets in Figure 5 show a linear $C(T)/T$ versus $T^2$ behavior for $2 \leq T \leq 15$ K. Then, it can be concluded that the Debye temperatures prepared under ambient pressure, 2 GPa, and 4 GPa are 504.4 K, 508.71 K and 515.36 K, respectively. The Debye temperature of the alloy prepared under high pressure increases. The contribution of lattice vibration to $C(T)$ of the Al–38wt.%Ni alloy at high temperature was obtained by fitting the $C(T)$ data over specified temperature ranges by the molar heat capacity. The fit of $C(T)$ over the temperature ranges from 15 K to 380 K and its extrapolation to higher temperatures is shown by the red solid curve in Figure 5.

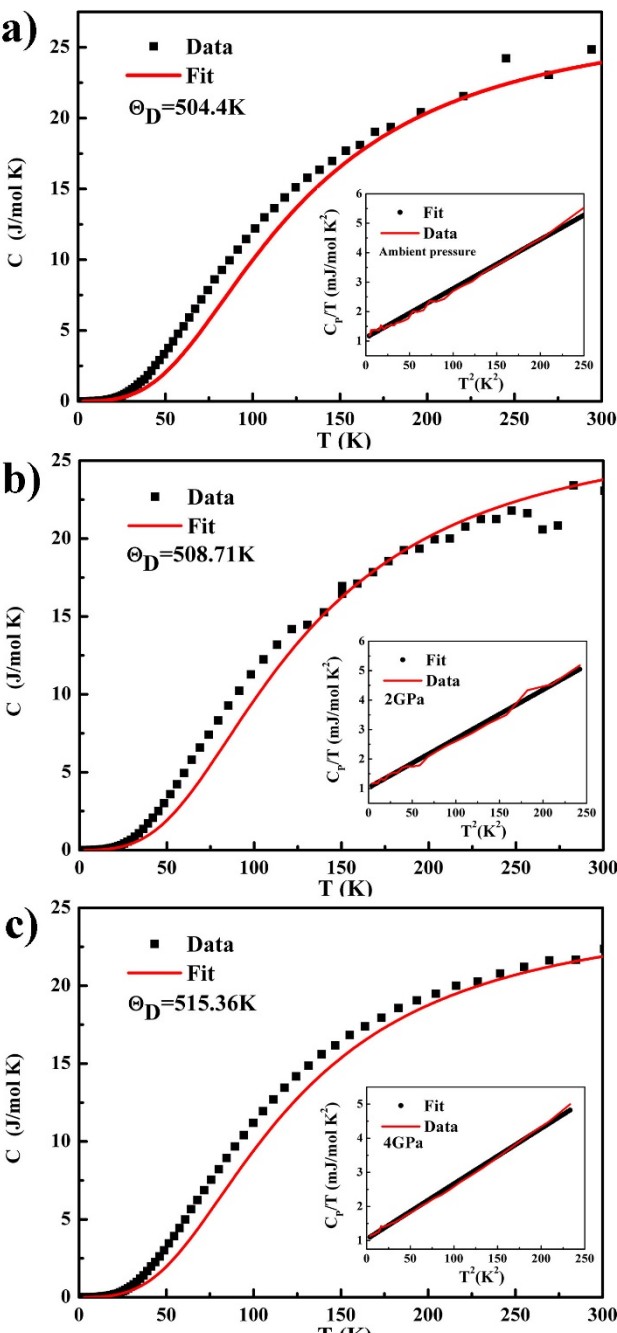

**Figure 5.** Heat capacity $C_V$ vs. temperature $T$ and fits of $C_{latt}(T)$ over restricted temperature intervals for (**a**) ambient pressure, (**b**) 2 GPa, (**c**) 4 GPa. The respective insets show $C_V(T)/T$ vs. $T^2$ for $2\ K < T < 15\ K$.

Table 3 shows the linear thermal expansion coefficients, binding energies, and bulk modulus of the Al–38wt.%Ni alloy under different pressures at room temperature. The calculation of binding energy is based on the model established by Mei [38], while the calculation of linear thermal expansion is based on the model build by Xu [39], and the bulk elastic modulus is calculated by means of molecular dynamics. The results show that the linear thermal-expansion coefficient decreases with the increase of pressure. However, the binding energy increases under high pressure, and the bulk modulus of elasticity shows the same trend.

**Table 3.** The linear-expansion coefficient, bond energy, and bulk modulus of the Al–38wt.%Ni alloy under different pressures at room temperature.

| Parameter | Ambient Pressure | 1 GPa | 2 GPa | 3 GPa | 4 GPa |
|---|---|---|---|---|---|
| Linear expansion coefficient ($\times 10^{-6}\,°\text{C}^{-1}$) | 19.98 | 9.56 | 8.98 | 8.48 | 8.05 |
| Bond energy (KJ/mol) | 261.0 | 274.0 | 287.0 | 380.1 | 313.2 |
| Bulk modulus (GPa) | 87.8371 | 94.794 | 101.68 | 108.57 | 115.38 |

By inserting the parameters as shown in Table 3 into the Equations (26)–(34), the potential energy curves of the Al–38wt.% Ni alloy under different pressures can be seen in Figure 6. It demonstrates that the potential energy in the lowest point decreases with the increase of pressure. And the potential curve shows a downward trend. When $r$ is less than the distance corresponding to the zero points of potential energy ($r_z$), the three curves are well coincident. When $r_z < r < r_0$, the three potential energy curves gradually separate, and the three potential energy curves are farthest away from each other at the lowest potential energy point. When $r > r_0$, the spacing of potential curves decreases gradually. Meanwhile, the three curves overlap again until $r > 5 \times 10^{-9}$ m. According to Equation (26), when $r = r_0$, the potential energy is equal to the binding energy. However, based on the parameters in Table 3, it can be seen that the bonding energy increases under high pressure. Finally, it proves the correctness of the potential energy curve.

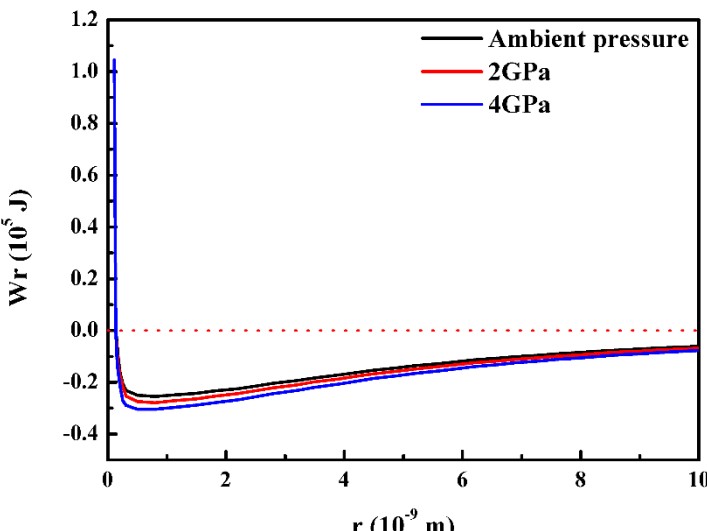

**Figure 6.** The potential curves of Al–38wt.%Ni alloys under different pressures.

## 6. Conclusions

The microstructure of the Al–38%Ni alloy solidified under different pressures was investigated, and the equation of solute segregation characteristics and stable growth wavelength of intermetallic compound Al$_3$Ni with nil solid solubility were established. Combined with the BCT dendrite growth model, the phase selection under pressure was also calculated. Finally, the potential was researched based on the low temperature specific heat-capacity curve. The following conclusions can be obtained:

(1) When the initial concentration of the Al–Ni alloy is lower than 25at.%Ni, no constitutional undercooling exists at the frontier of the Al$_3$Ni solid–liquid interface.

(2) Under the effect of high pressure, the interface temperature of the Al$_3$Ni$_2$ phase is lower than that of the Al$_3$Ni phase.

(3) The Debye temperatures of Al–38wt.%Ni alloys synthesized under ambient pressure, 2 GPa, and 4 GPa are 504.4 K, 508.71 K and 515.36 K, respectively. The potential energy in the lowest point decreases with the increase of pressure.

**Author Contributions:** Conceptualization, X.-H.W.; methodology, X.-H.W.; formal analysis, X.-H.Y.; investigation, X.-H.W. and D.D.; resources, X.-H.Y.; data curation, X.-H.W.; writing—original draft preparation, X.-H.W. and D.D.; writing—review and editing, X.-H.Y. All authors have read and agreed to the published version of the manuscript.

**Funding:** This work was supported by the Zhejiang Province Natural Science Foundation of China (Grant No.: LQ20E010003) and the National Natural Science Foundation of China [No. 52071165].

**Data Availability Statement:** All data, models, and code generated or used during the study appear in the submitted article.

**Conflicts of Interest:** The authors declare no conflict of interest.

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
