# Peer review of "Effect of High Pressure on the Solidification of Al–Ni Alloy"

_crystals, doi:10.3390/cryst11050478_

Round 1
Reviewer 1 Report
The work is prepared quite carefully, cites a wide range of good quality literature and contains both broad theoretical introduction and valuable experimental data. In my opinion, it is suitable for publication in a journal, after considering a number of comments. There are some minor scientific, linguistic and editorial mistakes:
- line 91- there is a bit of lack of motivation, why exactly 38%Ni composition was chosen?
- line 68 - Soldification->soldification
- line 86 - I suggest to add more theoretical background about Debye temperature. It is quite important for work but described with only 5 lines of Introduction, while phase transformations of different types of alloys has much more attention.
- line 100-110 - The description of the apparatus is somewhat inconsistent, I suggest keeping the notation Manufacturer, Model (City, State)
- line 178 - term "BSE" should be used after setting this abbreviation, like "backscattered electron (BSE) contrast in composition mode"
- line 185 - giving the results from EDS with an accuracy of 0.01% at is a bit of an abuse without specifying the measurement error. Does this accuracy matter? How many measuring points were made and how the statistical data was processed? Maybe it is enough to provide an error or standard deviation?
- line 192,195 - How the grain size wac calculated? Such irregular structure deserve more information.
- line 197 - figs 2 and 3 are hardly readable. In fig. 2 there are missing bars and a)b)... letters are totally invisible, while few symbols in Fig. 3 looks similiar
- line 200 - one peak (such tiny) is not a lot to clearly identify Al3Ni2. Is this theoretically the strongest peak for that phase?
- line 275 - what is the quality of the fit value? How to explain a imperfection of the data fit to the model?
- line 288 - in table all the pressure values are "1GPa"
- line 311 - What is the significance of the quoted conclusions for the development of new or similar alloys, what is the value of work beyond the new physical values?
Author Response
Dear Editor Agnieszka Rydz and Reviewers,
Thank you very much for your comments and explanation. We have carefully read your comments on the manuscript, titled “Effect of high pressure on the solidification of Al-Ni alloy” (Manuscript ID: crystals-1195477). We have revised the manuscript and responded point by point to the comments as listed below.
Response to Reviewer 1 Comments
The work is prepared quite carefully, cites a wide range of good quality literature and contains both broad theoretical introduction and valuable experimental data. In my opinion, it is suitable for publication in a journal, after considering a number of comments. There are some minor scientific, linguistic and editorial mistakes:
Point 1: line 91- there is a bit of lack of motivation, why exactly 38%Ni composition was chosen?
Response 1: we are really sorry and appreciate your suggestions. Following contents have been added.
As it contains both intermetallic compounds and peritectic reaction.
Point 2: line 68 - Soldification->soldification
Response 2: Thank you so much for your suggestion. And we appreciate your comments very much. It has been revised and marked in red.
Point 3: line 86 - I suggest to add more theoretical background about Debye temperature. It is quite important for work but described with only 5 lines of Introduction, while phase transformations of different types of alloys has much more attention.
Response 3: We really appreciate your comments. The introduction has been revised and marked in red.
Debye temperature is an important physical property of metal materials, it is a thermo-parameter to describe many physical phenomena of solid state materials which associated with the lattice vibrations[28]. It depends on properties such as elastic constants, lattice thermal conductivity and specifific heat [29]. Theoretically, Lindemann [30] suggested a relationship between Debye temperature and melting temperature based on the function of Einstein expressing the variation of the atomic heat of various elements. while Daoud [31] established a empirical expression about Debye temperature and bond length, Abrahams et al.[32] studied the relationships between Debye temperature and compressibility, microhardness etc.
Point 4: line 100-110 - The description of the apparatus is somewhat inconsistent, I suggest keeping the notation Manufacturer, Model (City, State)
Response 4: Thank you so much for your suggestions. The experiment part has been revised as follows:
Using pure Aluminum (99.99%) and Nickel (99.99%) as raw materials, 38 wt.% Al-Ni alloy was prepared by conventional casting technology. The high pressure solidification sample is a cylinder with a diameter of 6 mm and a length of 8 mm. The experiment was carried out on a 700 ton multi anvil device (Self assembled, Japan). The Bi phase transition was used as the fixed point to determine the pressure generated, and inserting hot junction of thermocouple (R-type) in the center of a heater was used to calibrate the temperature. The pressure is set at 2GPa and 4GPa. The sample is melted in a graphite tube furnace. Pyrophyllite is used as encapsulant and pressure transmitting medium. When the pressure raised to the target value for 50-60 minutes, the sample is kept at the target pressure and temperature for 1 hour to melt completely. Finally, sample was taken out for analysis after pressure relief. The morphology of the samples was obtained by SEM (Hitachi s-8100, Tokyo, Japan), and the composition was analyzed by EDS (Hitachi su8010, Tokyo, Japan). The low temperature heat capacity performance was tested with Physical Property Measurement System (PPMS-9, Quantum Design, US).
Point 5: line 178 - term "BSE" should be used after setting this abbreviation, like "Back-scattered electron (BSE) contrast in composition mode"
Response 5: Thank you so much and we really sorry for the mistakes we have made. Revised has been made and marked in red.
Point 6: line 185 - giving the results from EDS with an accuracy of 0.01% at is a bit of an abuse without specifying the measurement error. Does this accuracy matter? How many measuring points were made and how the statistical data was processed? Maybe it is enough to provide an error or standard deviation?
Response 6: We appreciate the reviewer’s comments. The standard sample has been produced to check the chemical composition. The ZAF coefficient and the error range have been provided in the figures below. It can be seen that the error ranges from 2.3% to 12.07%. The lower the element content, the greater the error. Especially, we just want to identify the Al3Ni2 (Ni 39.17at% ) and Al3Ni (Ni 25.01%) phase combined with the XRD results.
Point 7: line 192, 195 - How the grain size was calculated? Such irregular structure deserves more information.
Response 7: The grain size was calculated with the nano Measure software. Nano measure is a special software for size calibration and particle size distribution calculation of SEM and TEM images. For the grains with non-uniform shape, select from different directions, and select at least 30 grains to calculate the average value.
Point 8: line 197 - figs 2 and 3 are hardly readable. In fig. 2 there are missing bars and a)b)... letters are totally invisible, while few symbols in Fig. 3 looks similar.
Response 8: Once again, we are sorry and ashamed of such a mistake. And thank you so much for your help. Fig.2 and Fig.3 have been changed.
Point 9: line 200 - one peak (such tiny) is not a lot to clearly identify Al3Ni2. Is this theoretically the strongest peak for that phase?
Response 9: Theoretically, the peak intensity is proportional to the phase content, while the content of Al3Ni2 phase in the paper is relatively small. Therefore, the peak intensity will be lower.
Point 10: line 275 - what is the quality of the fit value? How to explain a imperfection of the data fit to the model?
Response 10: Thank you so much for your revision. And we appreciate it very much. However, When the R value is close to 1, the standard error value of the fitting parameter is more obvious, and the more stable the fitting result is. Meanwhile, fitting is based on all the data, so at both ends of the data, that is, the maximum and minimum values, because the data is not continuous, large error is normal.
Point 11: line 288 - in table all the pressure values are "1GPa"
Response 11: We are really sorry that we made such a stupid mistake. The table has been modified and marked in red.
Point 12: line 311 - What is the significance of the quoted conclusions for the development of new or similar alloys, what is the value of work beyond the new physical values?
Response 12: The first two conclusions are related to the redistribution of solute and the competitive growth of phases. For the last conclusion, The higher the Debye temperature, the greater the interatomic force, the smaller the coefficient of expansion and the greater the young's modulus.

Reviewer 2 Report
Authors investigate high-pressure high-temperature properties of well known Al-Ni system. High-pressure might change quite drastically phase composition as well as stability of metallic systems. As a result every high-pressure study of metallic systems has high importance for the community and relevant for further progress in the area. Manuscript is clear and give all needful information about the system. Nevertheless, before considering for the publication, several changes should be done. Authors discus quite intensive phase changes under ambient and high-pressure. I suggest to give a drawing of corresponding ambient pressure phase diagram and probably sketch of high-presssure changes in the phase diagram to givee better understanding of the phase evalutions under compression. It is not clear why 2 and 4 GPa as well as 504, 508 and 515 K were choosen for experimentation. I suggest to give a short ratoinal explanation for conditions choosen. Lines 97-98. Conventional casting should be described in more details. Materials and methods should also contain information about PXRD experiment desctibed n Fig. 3. How pressure and temperature were reliased? Was it quenching or slow cooling and decompression? How cooling and decompression influence the equilibration of the system? From the text it is not clear how decompressed samples represent high-pressure phases. Depending on decompression process, samples can equilibrate and represent again ambient pressure situation or can be quenched and represent high-pressure phases.Author Response
Dear Editor Agnieszka Rydz and Reviewers,
Thank you very much for your comments and explanation. We have carefully read your comments on the manuscript, titled “Effect of high pressure on the solidification of Al-Ni alloy” (Manuscript ID: crystals-1195477). We have revised the manuscript and responded point by point to the comments as listed below.
Response to Reviewer 2 Comments
Authors investigate high-pressure high-temperature properties of well known Al-Ni system. High-pressure might change quite drastically phase composition as well as stability of metallic systems. As a result every high-pressure study of metallic systems has high importance for the community and relevant for further progress in the area. Manuscript is clear and give all needful information about the system. Nevertheless, before considering for the publication, several changes should be done.
Point 1: Authors discuss quite intensive phase changes under ambient and high-pressure. I suggest to give a drawing of corresponding ambient pressure phase diagram and probably sketch of high-pressure changes in the phase diagram to give better understanding of the phase evaluations under compression.
Response: Thank you very much for your advice. It is our next research plan. Because only one component is not enough to plot the phase diagram at high pressure. Other Ni-Al alloys with high nickel content are being studied. It is believed that the phase diagram of Ni-Al alloy under high pressure will be obtained soon.
Point 2: It is not clear why 2 and 4 GPa as well as 504, 508 and 515 K were chosen for experimentation. I suggest to give a short rational explanation for conditions chosen.
Response: I'm really sorry, because of the limitation of the equipment in our team, the pressure can reach 4GPa at most. Therefore, we choose 2GPa and 4GPa in a limited range of conditions. Higher pressure, If necessary, we will seek cooperation with other research institutes or universities.
Point 3: Lines 97-98. Conventional casting should be described in more details. Materials and methods should also contain information about XRD experiment described in Fig. 3.
Response: We appreciate your comments. And the following changes have been made.
Using pure Aluminum (99.99%) and Nickel (99.99%) as raw materials, 38 wt.% Al-Ni alloy was prepared by button ingot melting furnace. The total weight of each button ingot was no more than 50g, and the diameter was about 40mm. And It was repeatedly melted for 5 times with properly current and power.
Point 4: How pressure and temperature were released? Was it quenching or slow cooling and decompression?
Response: The part 2 has been revised and marked in red as follows:
Using pure Aluminum (99.99%) and Nickel (99.99%) as raw materials, 38 wt.% Al-Ni alloy was prepared by button ingot melting furnace. The total weight of each button ingot was no more than 50g, and the diameter was about 40mm. And It was repeatedly melted for 5 times with properly current and power. The high pressure solidification sample is a cylinder with a diameter of 6 mm and a length of 8 mm. The experiment was carried out on a 700 ton multi anvil device (Self assembled, Japan). The Bi phase transition was used as the fixed point to determine the pressure generated, and inserting hot junction of thermocouple (R-type) in the center of a heater was used to calibrate the temperature. The pressure is set at 2GPa and 4GPa. The sample is melted in a graphite tube furnace. Pyrophyllite is used as encapsulant and pressure transmitting medium. When the pressure rises to the target value, the sample is kept at the target pressure and temperature for 1 hour to ensure complete melting. Then quickly reduce the current to 0, stop heating and make the sample cool and solidify. Finally, samples were taken out for analysis after pressure relief. The morphology of the samples was obtained by SEM(Hitachi s-8100, Tokyo, Japan), and the composition was analyzed by EDS (Hitachi su8010, Tokyo, Japan). The low temperature heat capacity performance was tested with Physical Property Measurement System (PPMS-9, Quantum Design, US).
Point 5: How cooling and decompression influence the equilibration of the system? From the text it is not clear how decompressed samples represent high-pressure phases. Depending on decompression process, samples can equilibrate and represent again ambient pressure situation or can be quenched and represent high-pressure phases.
Response: Thank you very much for your comments. We are very sorry that we didn't explain the process of high pressure experiment in our first writing. It can be seen from the revised content that the high-pressure experiment is a process of raise the pressure → melt the sample by heating→holding temperature and pressure for 1h→cooling and solidification → pressure relief. It can be seen that the sample is solidified under the action of pressure, so the phase generated under pressure can be preserved.
